# IBGS: Image-Based Gaussian Splatting

**Hoang Chuong Nguyen**[1]    **Wei Mao**    **Jose M. Alvarez**[2]    **Miaomiao Liu**[1]

[1]Australian National University        [2]NVIDIA

hoangchuong.nguyen@anu.edu.au   miaomiao.liu@anu.edu.au
wei.mao.research@gmail.com   josea@nvidia.com

## Abstract

3D Gaussian Splatting (3DGS) has recently emerged as a fast, high-quality method for novel view synthesis (NVS). However, its use of low-degree spherical harmonics limits its ability to capture spatially varying color and view-dependent effects such as specular highlights. Existing works augment Gaussians with either a global texture map, which struggles with complex scenes, or per-Gaussian texture maps, which introduces high storage overhead. We propose Image-Based Gaussian Splatting, an efficient alternative that leverages high-resolution source images for fine details and view-specific color modeling. Specifically, we model each pixel color as a combination of a base color from standard 3DGS rendering and a learned residual inferred from neighboring training images. This promotes accurate surface alignment and enables rendering images of high-frequency details and accurate view-dependent effects. Experiments on standard NVS benchmarks show that our method significantly outperforms prior Gaussian Splatting approaches in rendering quality, without increasing the storage footprint. Our project page is available at https://hoangchuongnguyen.github.io/ibgs.

## 1   Introduction

Recently, Neural Radiance Fields (NeRF) [23] and 3D Gaussian Splatting (3DGS) [16] have emerged as advanced techniques for Novel View Synthesis (NVS) thanks to their high-quality image rendering. 3DGS, in particular, outperforms NeRF in terms of rendering speed and faster optimization. However, since each Gaussian primitive in 3DGS can only represent a single color at a given camera viewpoint, 3DGS struggles to recover high-frequency details of the scene appearance without a large number of Gaussians [4]. Furthermore, due to the smooth characteristic of the color representation i.e., spherical harmonic (SH) functions, it is hard for 3DGS to capture complex view-dependent effects such as reflections and specular highlights [4].

To address this issue, recent works have attempted to model Gaussian's spatially varying colors by either mapping each Gaussian-ray intersection to a global texture map [34], or a per-Gaussian texture map [4, 28, 33]. While global texture maps perform well for single-object scenes [34], they struggle with complex multi-object scenes due to the difficulty of learning a global mapping. Per-Gaussian texture map [4, 28, 33] can handle real-world scenes with multiple objects, but it incurs a significant storage overhead because the number of parameters per Gaussian grows quadratically as the texture-map resolution increases. This storage overhead constrains the resolution of per-Gaussian's texture map, leading to inferior modeling of high-frequency details in the rendered images. Additionally, such texture map still cannot handle view-dependent effects.

In this work, we propose a drastically different approach to render high-frequency details and handle view-dependent effects while avoid significantly increasing the storage memory. Specifically, inspired by image-based rendering techniques [8, 22], we introduce an Image-Based Gaussian Splatting (IBGS) approach that utilizes the high-frequency details and view-dependent effects captured in training images. During rendering, the color of a pixel consists of two components i.e., the base color

from the SH functions following the standard rasterization process of 3DGS, and the residual color learnt from the corresponding pixel intensities of neighboring training images. The base color is used to handle most surface appearance while the color residual augments the base color with fine-grained details and view-dependent effects which are preserved in the training images. To model the color residuals, we propose a novel color residual prediction module. Specifically, for each ray/pixel, we first project the intersection points between the ray and Gaussians onto neighboring training images to obtain pixel colors, which are then aggregated to get warped colors. Then, the warped colors together with the base colors are processed by a lightweight neural network to predict the color residual for each pixel. We further introduce an image synthesis loss that leverages those warped colors, enforcing both geometric accuracy and image quality. This leads to more precise Gaussian parameters with high opacity centered around the true surface, allowing us to prune more Gaussians of low opacity while maintaining the rendering quality.

Furthermore, leveraging neighboring views enables our method to address inconsistent exposure across training views caused by the auto-exposure behavior of modern cameras. Different from prior work [6, 17] that jointly optimize an affine transformation matrix for each training view, we assume that images taken at nearby locations share similar global lighting conditions, and thus correct the camera exposure at novel views by mimicking the exposure setting of the closest source view. Unlike existing works [6, 17] that only correct the exposure at training views, our strategy can generalize to images rendered at any viewpoint.

In summary, our contributions are: i) We propose an image-based Gaussian Splatting pipeline that captures both high-frequency details and view-dependent effects that are challenging for prior methods to address. ii) We introduce a color residual module that leverages the training images to obtain better rendering quality with less number of Gaussians. iii) We introduce an exposure correction strategy, helping to improve the brightness of images rendered at any viewpoint by mimicking the exposure settings of their nearest neighbouring view. Our method sets a new state-of-the-art performance on three benchmark datasets: Tanks and Temples, Deep Blending, and Mip-NeRF360.

## 2 Related Works

**Image-based rendering** aims to generate novel views by "borrowing" pixels from a set of source images. The target pixel is a weighted blending of corresponding pixels obtained from those images. In early works [8, 11, 22], such blending weights are computed based on ray distance [22] or scene geometry [8]. Other works either tried to improve the scene geometry [5, 13] or use optical flow for better correspondence [1, 3, 9]. More recently, with the advance of neural rendering techniques [23], researchers have explored integrating it with image-based rendering [29, 31]. In particular, to render a target pixel/ray, Suhail et al. [29] first finds the corresponding epipolar lines in source views and sample points along such lines to obtain color features. These features are further fed into two feature aggregation modules subsequently for the final color. IBRNet [31] follows the volume rendering process as in NeRF [23]. The color and densities of the sampled points on target ray are computed by a transformer [30] with the features from source views as input. Despite their impressive results, their rendering is time-consuming due to the use of large feature aggregation networks (i.e., the transformers) and uniform sampling along the ray. By contrast, to the best of our knowledge, we propose the first image-based Gaussian splatting method that not only obtains fine-grained details from source images but also maintain fast rendering. Thanks to 3DGS, we only require projecting intersection points of ray with Gaussians, which are fairly sparse, to source views for aggregating image features. Moreover, instead of directly learning the final color from a large network, we propose to learn a residual to the base color, which only requires a lightweight network i.e., a nine-layer convolutional network with $3 \times 3$ with kernels.

**Gaussian Splatting.** 3DGS [16] renders images at novel-views by performing alpha blending of the Gaussians color splatted onto the image plane. Although each splatted Gaussian can have a large extent, its color is shared across all pixels, making it challenging for 3DGS to reconstruct fine-grained textures without using many Gaussians. To address this problem, prior works [4, 28, 33, 34] attempt to model Gaussian spatially varying color by learning a mapping from each Gaussian-ray intersection to a texture map. However, while learning a global texture map is challenging in scenes having multiple objects, learning per-Gaussian texture maps [4, 28, 33] leads to higher storage requirement since it requires to store texture maps of all Gaussians. Additionally, these methods struggle in recovering complex view-dependent color as they utilize low-degree SH functions which have limited capacity

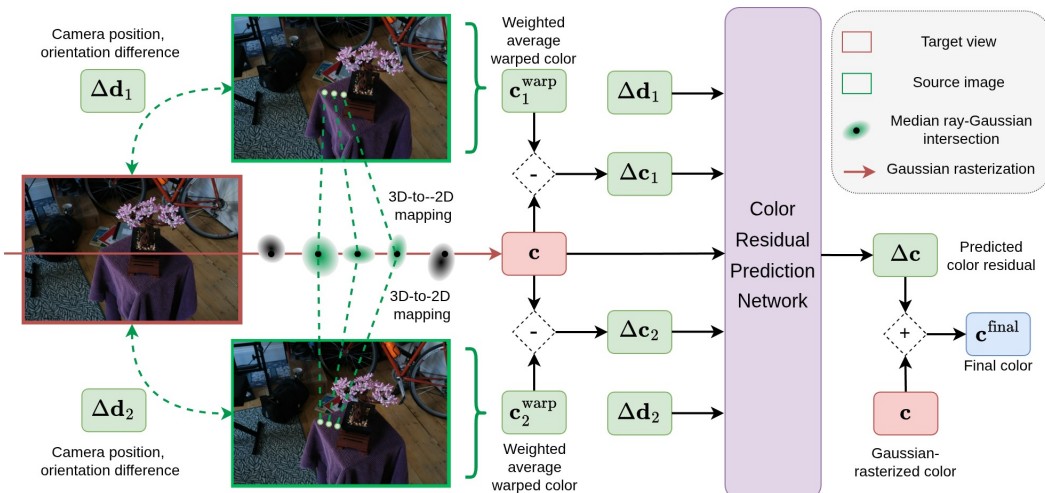

Figure 1: Our pipeline. The color of each pixel ($\mathbf{c}^{\text{final}}$) consists of two components: a base color $\mathbf{c}$ (in pink boxes) which follows the standard 3DGS rendering process and a color residual $\Delta\mathbf{c}$ predicted from the warped color of different source views $\mathbf{c}_m^{\text{warp}}$. While the figure shows an example of using only two source views, in practice, our method can process an arbitrary number of source views.

to handle complex view-dependent colors [4]. Unlike these methods, we leverage information from training views to render photorealistic images by predicting high residuals for pixels whose color cannot be fully recovered via Gaussian rasterization (i.e., the base color), especially in regions with fine-grained details or view-dependent colors.

Apart from color modeling, existing works also target improving other aspects of Gaussian Splatting. In particular, [7, 12, 15] pursue more accurate 3D reconstruction by enforcing flat Gaussians via a hard [7, 15] or soft constraint [12]. On the other hand, [10, 25, 26] propose compression and quantization methods to reduce the memory requirements for storing optimized Gaussians while preserving rendering quality. [35] improves the adaptive density control strategy of 3DGS by exploiting the per-pixel gradient directions, whereas [18] views the Gaussian densification process as state transition of Markov Chain Monte Carlo samples. Our method is orthogonal to these works and can be integrated with them to further boost the performance or reduce memory usage.

## 3 Method

### 3.1 Preliminary: 3D Gaussian Splatting

3DGS [16] represents the scene as a set of 3D Gaussian primitives. Each 3D Gaussian $\mathcal{G}$ is parameterized by a 3D position $\boldsymbol{\mu} \in \mathbb{R}^3$ and a covariance matrix $\boldsymbol{\Sigma} \in \mathbb{R}^{3\times3}$, where the covariance is decomposed into a rotation matrix $\mathbf{R} \in \mathbb{SO}(3)$ and a diagonal scale matrix $\mathbf{S} \in \mathbb{R}^{3\times3}$ such that $\boldsymbol{\Sigma} = \mathbf{R}\mathbf{S}\mathbf{S}^T\mathbf{R}^T$. To render an image at a viewpoint, each 3D Gaussian $\mathcal{G}$ is splatted on the image plane to obtain a 2D Gaussian $\mathcal{G}^{\text{2D}}$. The color of a pixel $\mathbf{p}$ can be then computed using the volume rendering equation,

$$\mathbf{c}(\mathbf{p}) = \sum_{i=1}^{K} w_i\, \mathbf{c}_i = \sum_{i=1}^{K} w_i\, \Psi_l(\mathbf{h}_i, \mathbf{v}_i), \qquad \text{where} \quad w_i = \alpha_i T_i, \quad \text{and} \quad T_i = \prod_{j=1}^{i-1}(1 - \alpha_j), \quad (1)$$

with $\mathbf{v}_i$ denoting the vector from the camera center to the center of the $i^{\text{th}}$ Gaussian, $\Psi_l(\mathbf{h}, \mathbf{v})$ mapping the SH coefficients $\mathbf{h}$ of the Gaussian to a color $\mathbf{c}$, conditioned on the direction $\mathbf{v}$. The scalar $l \in \mathbb{R}$ indicates the SH degree, which determines the expressivity of $\Psi_l(\mathbf{h}, \mathbf{v})$. The weight $\alpha_i = o_i \mathcal{G}_i^{\text{2D}}(\mathbf{p})$ is defined as the product of the Gaussian opacity $o_i$ and 2D Gaussian value evaluated at pixel $\mathbf{p}$.

During training, the attributes of each Gaussian, including (1) its colors represented using spherical harmonic coefficients $\mathbf{h} \in \mathbb{R}^{3(l+1)^2}$, (2) the opacity $o \in \mathbb{R}$, (3) the center $\boldsymbol{\mu} \in \mathbb{R}^3$, (4) the rotation matrix parameterized by a quaternion $\mathbf{q} \in \mathbb{R}^4$, (5) scaling factors $\mathbf{s} \in \mathbb{R}^3$ are optimized using the color rendering loss.

## 3.2 Modeling Spatially Varying and View-Dependent Color

In the color modeling of 3DGS [16], at a given viewpoint, although a Gaussian $\mathcal{G}_i$ can cover multiple pixels in the image, it represents only a single color, as the SH coefficients $\mathbf{h}$ and view direction $\mathbf{v}$ are shared across all pixels. This limits the model's capacity in modeling color of regions with high-frequency details. Moreover, due to the low-degree SH function ($l \leq 3$) utilized by 3DGS [16] to model view-dependent color, this method struggles to capture complex view-dependent effects, such as reflections or specular highlights. A naive solution is to increase the SH degree $l$, which quadratically increases the number of SH coefficients, thereby leading to high storage requirement.

In this work, we present a solution to model high-frequency details and view-dependent color of the image without increasing the storage footprint. Specifically, we propose IBGS, an image-based Gaussian splatting method that models (1) the base color from SH functions, (2) a color residual term capturing view-specific and high-frequency information from neighboring source images. We formulate it as follows:

$$\mathbf{c}^{\text{final}}(\mathbf{p}) = \underbrace{\sum_{i=1}^{N} w_i \, \Psi_l(\mathbf{h}_i, \mathbf{v}_i)}_{\text{Base color } \mathbf{c}(\mathbf{p})} + \underbrace{\mathcal{F}\Big(\mathbf{c}(\mathbf{p}), \, \mathbf{d}(\mathbf{p}), \, \{\Delta \mathbf{c}_m(\mathbf{p})\}_{m=1}^{M}, \, \{\Delta \mathbf{d}_m(\mathbf{p})\}_{m=1}^{M}\Big)}_{\text{Color residual } \Delta \mathbf{c}(\mathbf{p})}, \quad (2)$$

where $\mathbf{d}(\mathbf{p}) \in \mathbb{R}^3$ is the direction of the camera ray passing through pixel $\mathbf{p}$. $\Delta \mathbf{c}_m \in \mathbb{R}^3$ and $\Delta \mathbf{d}_m \in \mathbb{R}^4$ denotes appearance and camera features extracted from the $m^{\text{th}}$ nearby source view, respectively. $\mathcal{F}(\cdot)$ is a lightweight network that takes the extracted features as input to predict a residual term $\Delta \mathbf{c}(\mathbf{p})$ supplementing the base color, $\mathbf{c}(\mathbf{p})$, produced by Gaussian rasterization.

Given the extracted multi-view features $\Delta \mathbf{c}_m$ that capture high-frequency details and color variation across different viewpoints, we use this features to predict the pixel color in the current view. By utilizing the multi-view color observations, our model learns how lighting effect changes across viewpoint, thus being able to produce accurate view-dependent color for the target view.

In the following sections, we first describe the two main components of our method: feature extraction from source views (Sec. 3.3) and color residual prediction (Sec. 3.4). Then, we present an exposure correction approach (Section. 3.5) to correct the exposure of the rendered images caused by inconsistent exposure camera setting. Fig. 1 depicts the overall pipeline of our method.

## 3.3 Feature Extraction from Source Views

To predict color residual, $\Delta \mathbf{c}(\mathbf{p})$, we first extract features from multiple source images. For a target pixel $\mathbf{p}$, we obtain the color information $\Delta \mathbf{c}_m(\mathbf{p})$ from each source image by first finding the intersection between (1) the camera ray originating at the camera center $\mathbf{o}$ with direction $\mathbf{d}(\mathbf{p})$ and (2) the plane parameterized by the Gaussian center $\boldsymbol{\mu}_i$ and its normal vector $\mathbf{n}_i$,

$$\mathbf{x}_i(\mathbf{p}) = \mathbf{o} + \frac{\mathbf{n}_i^T(\boldsymbol{\mu}_i - \mathbf{o})}{\mathbf{n}_i^T \mathbf{d}(\mathbf{p})} \mathbf{d}(\mathbf{p}). \quad (3)$$

Here we incorporate a normal vector $\mathbf{n}_i \in \mathbb{R}^3$ as an additional learnable attribute of each Gaussian $\mathcal{G}_i$. Given the intersection point $\mathbf{x}_i(\mathbf{p})$, we project it onto the image plane of nearby source views from which the color information is extracted. For the $m^{\text{th}}$ source view, this is achieved as,

$$\mathbf{c}_{i,m}^{\text{warp}}(\mathbf{p}) = \mathcal{B}(\pi_m(\mathbf{x}_i(\mathbf{p})), \mathbf{C}_m^{\text{real}}) \quad (4)$$

with $\pi_m(\mathbf{x})$ denoting a function that projects the intersection point to the image plane of the source view. The function $\mathcal{B}(\cdot)$ takes an image coordinate as input to produce bilinear-interpolated color obtained from the input source image $\mathbf{C}_m^{\text{real}} \in \mathbb{R}^{H \times W \times 3}$.

Based on Eq. 4, the warped color $\mathbf{c}_{i,m}^{\text{warp}}(\mathbf{p})$ is only accurate if the intersection point $\mathbf{x}_i(\mathbf{p})$ is close to the actual surface. This implies that it is not necessary to project all the Gaussian-ray intersections to the source views, as floating Gaussians that are far from true surface introduce noise into the extracted appearance features. Following 2DGS [15], we consider the actual surface lies near the median intersections such that the accumulated transmittance $T_i$ (see Eq. 1) is close to 0.5, and thus only project the $K$ median intersection points $\{\mathbf{x}_{k,m}\}_{k=1}^{K}$ to the source views. As a result, we obtain a set of $K$ warped colors $\{\mathbf{c}_{k,m}^{\text{warp}}\}_{k=1}^{K}$ for each source view $m$.

Subsequently, we compute the weighted average color for each source view and measure its deviation from the Gaussian-rasterized color $\mathbf{c}(\mathbf{p})$ (computed via Eq. 1) as below,

$$\Delta\mathbf{c}_m(\mathbf{p}) = \mathbf{c}_m^{\text{warp}}(\mathbf{p}) - \mathbf{c}(\mathbf{p}), \qquad \mathbf{c}_m^{\text{warp}}(\mathbf{p}) = \sum_{k=1}^{K} \frac{w_k\, \mathbf{c}_{k,m}^{\text{warp}}(\mathbf{p})}{\sum_{k=1}^{K} w_k}, \qquad (5)$$

where $w_k$ is the same blending weight of the Gaussian colors computed from Eq. 1. Intuitively, $\mathbf{c}_m^{\text{warp}}(\mathbf{p})$ is used to approximate the true pixel color by leveraging the information from neighboring views. Thus, it constrains the weight $w_k$ of the Gaussians near the true surface to be larger than those of others. Apart from the appearance features, we also compute $\Delta\mathbf{d}_m$, the difference in the camera position and orientation between the target and each source view,

$$\Delta\mathbf{d}_m = \begin{bmatrix} \mathbf{o}_m - \mathbf{o} \\ \mathbf{d}_m(\mathbf{p})^T\mathbf{d}(\mathbf{p}) \end{bmatrix}, \qquad \mathbf{d}_m(\mathbf{p}) = \frac{\mathbf{x}(\mathbf{p}) - \mathbf{o}_m}{||\mathbf{x}(\mathbf{p}) - \mathbf{o}_m||_2}, \qquad \mathbf{x}(\mathbf{p}) = \sum_{k=1}^{K} \frac{w_k\, \mathbf{x}_k(\mathbf{p})}{\sum_{k=1}^{K} w_k}, \qquad (6)$$

with $\mathbf{o}_m$ being the camera center of the $m^{\text{th}}$ source view. We repeat this feature extraction process for the $M$ nearby source views, yielding a set of color features $\{\Delta\mathbf{c}_m\}_{m=1}^{M}$ and camera features $\{\Delta\mathbf{d}_m\}_{m=1}^{M}$ which are used as input to the color residual prediction network.

## 3.4 Color Residual Prediction

We employ a lightweight network to predict the color residuals, consisting of two main components: a per-pixel feature extractor $\mathcal{E}(\cdot)$ and a CNN decoder $\mathcal{D}(\cdot)$. The extractor has a PointNet-style structure [27] to handle an arbitrary number of source views $M$. For each view $m$, it processes the color feature $\Delta\mathbf{c}_m(\mathbf{p})$ and camera features $\Delta\mathbf{d}_m(\mathbf{p})$ through two linear layers of 32 output dimension followed by ReLU activation function,

$$\mathbf{f}_m(\mathbf{p}) = \mathcal{E}\big(\Delta\mathbf{c}_m(\mathbf{p}),\, \Delta\mathbf{d}_m(\mathbf{p})\big). \qquad (7)$$

We then apply max-pooling to the set of vectors $\{\mathbf{f}_m(\mathbf{p})\}_{m=1}^{M}$ to obtain the aggregated feature $\bar{\mathbf{f}}(\mathbf{p}) \in \mathbb{R}^{32}$. Assembling these features across all pixels yields a feature map $\mathbf{F} \in \mathbb{R}^{H \times W \times 32}$. Similarly, by stacking $\mathbf{c}(\mathbf{p})$ and $\mathbf{d}(\mathbf{p})$ over all pixels, we form the Gaussian-rasterized image $\mathbf{C} \in \mathbb{R}^{H \times W \times 3}$ and the ray-direction map $\mathbf{D} \in \mathbb{R}^{H \times W \times 3}$, respectively. This information are then fed into a nine-layer convolutional decoder (with kernel size of 3) to predict the color residual map $\Delta\mathbf{C} \in \mathbb{R}^{H \times W \times 3}$.

$$\Delta\mathbf{C} = \mathcal{D}(\mathbf{C}, \mathbf{D}, \mathbf{F}). \qquad (8)$$

Finally, we obtain the final image by adding the predicted residuals to the Gaussian-rasterized image,

$$\mathbf{C}^{\text{final}} = \mathbf{C} + \Delta\mathbf{C} \qquad (9)$$

## 3.5 Exposure Correction

Due to varying lighting conditions at different locations, cameras with an auto-exposure setting can capture images with inconsistent brightness, which introduces noise into the optimization of the Gaussians. To stabilize the training, prior works [6, 17] optimize an color affine transformation matrix for each training view. This approach, however, can not generalize to correct the exposure of images rendered at novel views. To address this issue, we assume that images taken at nearby locations share similar global lighting conditions and thus propose to correct the exposure of the Gaussian-rasterized image by mimicking the exposure setting of the closest source view. In particular, we first obtain an affine transformation matrix $\mathbf{A}^{\star}$ representing the exposure at the target view by solving the following least-square problem,

$$\mathbf{A}^{\star} = \arg\min_{\mathbf{A} \in \mathbb{R}^{3 \times 4}} \sum_{\mathbf{p} \in \chi} \left\| \mathbf{A} \begin{bmatrix} \mathbf{c}(\mathbf{p}) \\ 1 \end{bmatrix} - \mathbf{c}_1^{\text{warp}}(\mathbf{p}) \right\|_2^2 \qquad (10)$$

where $\chi$ is a set of pixels with valid coordinate when mapping to the source view, and $\mathbf{c}_1^{\text{warp}}(\mathbf{p})$ is color warped from the nearest source view. After that, we use $\mathbf{A}^{\star}$ to correct the exposure of the rendered image as, $\mathbf{c}^{\text{expo}}(\mathbf{p}) = \mathbf{A}^{\star} \begin{bmatrix} \mathbf{c}(\mathbf{p}) \\ 1 \end{bmatrix}$. Note that if exposure correction is applied, the color with corrected exposure $\mathbf{c}^{\text{expo}}(\mathbf{p})$ is used in place of the originally rendered color $\mathbf{c}(\mathbf{p})$ for computing appearance features (Eq. 5), residual prediction (Eq. 8) and obtaining the final image (Eq. 9).

### 3.6 Optimization

The overall loss function used to train our method is,

$$\mathcal{L} = \mathcal{L}_{\text{rgb}} + \lambda_1 \mathcal{L}_{\text{photo}} + \lambda_2 \mathcal{L}_{\text{normal}} \tag{11}$$

where $\lambda_1$, $\lambda_2$ are loss weights.

**Color Rendering Loss $\mathcal{L}_{\text{rgb}}$.** We compute the loss for both the final image and the Gaussian-rasterized image (i.e., the base image). The two terms are balanced by the weight $\gamma$ as follows,

$$\mathcal{L}_{\text{rgb}} = \gamma \mathbb{L}(\mathbf{C}, \mathbf{C}^{\text{real}}) + (1 - \gamma)\mathbb{L}(\mathbf{C}^{\text{final}}, \mathbf{C}^{\text{real}}). \tag{12}$$

In particular, $\mathbb{L}$ is defined as

$$\mathbb{L}(\mathbf{C}, \mathbf{C}^{\text{gt}}) = \beta \mathcal{L}_1(\mathbf{C}, \mathbf{C}^{\text{gt}}) + (1 - \beta)\mathcal{L}_{\text{SSIM}}(\mathbf{C}, \mathbf{C}^{\text{gt}}). \tag{13}$$

where $\beta$ is set to 0.8 and $\mathbf{C}, \mathbf{C}^{\text{gt}} \in \mathbb{R}^{H \times W \times 3}$ are the rendered and ground-truth image, respectively.

**Multi-view Color Consistency Loss $\mathcal{L}_{\text{photo}}$.** We also enforce photometric consistency across neighboring views to encourage accurate pixel matching,

$$\mathcal{L}_{\text{photo}} = \frac{1}{M} \sum_{m=1}^{M} \mathbb{L}(\mathbf{C}_m^{\text{warp}}, \mathbf{C}^{\text{real}}), \tag{14}$$

where $\mathbf{C}_m^{\text{warp}}$ is the warped image obtained by stacking all $\mathbf{c}_m^{\text{warp}}(\mathbf{p})$ computed in Eq. 5.

**Normal Consistency Loss $\mathcal{L}_{\text{normal}}$.** Following 2DGS [15], we apply the normal consistency loss to improve the overall geometry,

$$\mathcal{L}_{\text{normal}} = \frac{1}{|\Omega|} \sum_{\mathbf{p} \in \Omega} (1 - \mathbf{N}(\mathbf{p})^T \mathbf{N}_{\text{depth}}(\mathbf{p})) \tag{15}$$

with $\Omega$ being a set of all pixel coordinates and $\mathbf{N}$ denoting the rasterized normal map. $\mathbf{N}_{\text{depth}}$ is the normal map derived via finite difference of the point map $\mathbf{X}$ constructed from $\mathbf{x}(\mathbf{p})$ (see Eq. 6).

**Visibility-based Source Views Selection.** To find the nearby source views, we first compute the distance between the target and each source view, then use the closest $S$ views as candidates to search for $M$ visible source views (i.e., $M \leq S$). Specifically, for each pixel $\mathbf{p}$, we only use the $s^{\text{th}}$ source view for feature extraction if it satisfies the following the condition,

$$\frac{|z(\mathbf{x}(\mathbf{p})) - z(\mathbf{x}_s^{\text{warp}}(\mathbf{p}))|}{z(\mathbf{x}(\mathbf{p})) + z(\mathbf{x}_s^{\text{warp}}(\mathbf{p}))} \leq \tau, \qquad \text{with} \quad \mathbf{x}_s^{\text{warp}}(\mathbf{p}) = \mathcal{B}(\pi_s(\mathbf{x}(\mathbf{p})), \tilde{\mathbf{X}}_s) \tag{16}$$

where $\tau$ is a depth error threshold, $z(\mathbf{x})$ denotes the depth value of the 3D point $\mathbf{x}$. The point map $\tilde{\mathbf{X}}_s$ can be obtained by transforming the point map $\mathbf{X}_s$ of source view to the coordinate system of the target view. Intuitively, this approach performs depth consistency check to exclude the source views in which the point $\mathbf{x}(\mathbf{p})$ is not visible.

## 4 Experiments

### 4.1 Experimental Setup

**Dataset.** Following 3DGS [16], we evaluate the NVS performance of our method using 2 scenes in the Tanks and Temples (TNT) [20] dataset, 2 scenes in the Deep Blending [14] dataset, and 9 scenes in the Mip-NeRF360 [2] dataset. We also show the results on 3 scenes in the Shiny dataset [32] which pose challenging view-dependent effects including specular highlight, reflection and disc diffraction. For all scenes, we use every $8^{\text{th}}$ image for evaluation, and the rest for training.

**Implementation Details.** Similar to 3DGS [16], we train our method for $30,000$ iterations. During the first $7,000$ iterations, we set $\lambda_1 = \lambda_2 = 0$ and only activate the photometric and normal consistency loss thereafter with $\lambda_1 = 0.3$ and $\lambda_2 = 0.03$. The weight $\gamma$ is initially set to 1, and then decreased to 0.5 during the last $20,000$ iterations. Regarding hyper-parameters, we set SH degree $l = 2$, number of median intersection points $K = 4$, number of candidate source views $S = 4$, number of visible source views $M = 3$ and depth error threshold $\tau = 0.001$. We also prune Gaussians with opacity lower than 0.05. Following [6], we apply the exposure compensation from [6] and our proposed exposure correction method only to the TNT dataset. We use Adam optimizer [19] to train the residual prediction network. The initial learning rate is 0.001, which halves at iterations 18,000 and 25,000. All experiments are conducted using a single RTX 4090 GPU.

Table 1: Comparison between our method and previous works in three datasets. We measure the storage memory (Mem) in MB and the number of Gaussians (#Gauss) in millions. We report two released result from TexturedGaussian [4], with and without total memory usage.

| Dataset | Mip-NeRF 360 | | | | | Tanks&Temples (TNT) | | | | | Deep blending | | | | |
|---|---|---|---|---|---|---|---|---|---|---|---|---|---|---|---|
| Method \| Metric | PSNR↑ | SSIM↑ | LPIPS↓ | #Gauss | Mem | PSNR | SSIM | LPIPS | #Gauss | Mem | PSNR | SSIM | LPIPS | #Gauss | Mem |
| Mip-NeRF 360 [2] | 27.69 | 0.792 | 0.237 | × | 8.6 | 22.22 | 0.759 | 0.257 | × | 8.6 | 29.40 | 0.901 | 0.245 | × | 8.6 |
| Instant-NGP [24] | 25.30 | 0.671 | 0.371 | × | 13 | 21.72 | 0.723 | 0.330 | × | 13 | 23.62 | 0.797 | 0.423 | × | 13 |
| 3DGS [16] | 27.69 | 0.825 | 0.203 | 3.22 | 764 | 23.11 | 0.840 | 0.184 | 1.75 | 415 | 29.53 | 0.904 | 0.242 | 3.14 | 745 |
| SuperGauss [33] | 27.31 | 0.815 | 0.209 | 3.04 | 1021 | 23.72 | 0.847 | 0.179 | 1.50 | 502 | 28.83 | 0.901 | 0.250 | 2.27 | 762 |
| TexturedGauss [4] | 27.35 | 0.827 | 0.186 | – | – | 24.26 | 0.854 | 0.168 | – | – | 28.33 | 0.891 | 0.270 | – | – |
| TexturedGauss* [4] | 27.26 | – | – | 3.50 | 1047 | 24.28 | – | – | 1.30 | 691 | 28.52 | – | – | 1.00 | 668 |
| Ours | **28.33** | **0.837** | **0.186** | 1.59 | 291 | **24.84** | **0.869** | **0.148** | 0.75 | 143 | **30.12** | **0.912** | **0.237** | 1.11 | 197 |

Table 2: Comparison on three scenes in the Shiny dataset with challenging view-dependent colors.

| Scene (effect) | Guitars (specular highlight) | | | | | Lab (reflection) | | | | | CD (diffraction) | | | | |
|---|---|---|---|---|---|---|---|---|---|---|---|---|---|---|---|
| Method\|Metric | PSNR↑ | SSIM↑ | LPIPS↓ | #Gauss | Mem | PSNR | SSIM | LPIPS | #Gauss | Mem | PSNR | SSIM | LPIPS | #Gauss | Mem |
| 3DGS [16] | 29.37 | 0.947 | 0.131 | 0.41 | 97 | 29.17 | 0.927 | 0.123 | 0.63 | 150 | 29.10 | 0.935 | 0.110 | 0.51 | 121 |
| SuperGauss [33] | 30.43 | 0.952 | 0.121 | 0.39 | 131 | 29.38 | 0.932 | 0.107 | 0.61 | 204 | 29.49 | 0.944 | 0.091 | 0.70 | 234 |
| Ours | **35.65** | **0.953** | **0.105** | 0.18 | 46 | **35.06** | **0.966** | **0.056** | 0.27 | 66 | **35.23** | **0.955** | **0.060** | 0.27 | 69 |

## 4.2 Results

In Tab. 1, we show the comparison of our method with prior methods in terms of NVS performance, number of Gaussians and storage usage. The results reveal that our method consistently achieve the best image quality across all datasets. For the PSNR metrics, our method gains an improvement of at least **0.64**, **0.56** and **0.59** dB in the Mip-NeRF 360, TNT and Deep blending datasets, respectively. Notably, on the Mip-NeRF 360 and TNT datasets, our method reduces the number of Gaussians and the storage by at least **62%** and **42%**, respectively, compared to existing Gaussian Splatting methods [4, 16, 33] and still outperforms them. For the Deep blending dataset, although we use slightly more Gaussians than TexturedGauss [4], our method consumes **70%** less storage. This is because TexturedGauss needs to store the texture maps of all Gaussians, which requires significantly more memory compared to storing the source images as in our method.

Tab. 2 presents the comparisons on three scenes in the Shiny [32] dataset with challenging view-dependent effects. For this dataset, we train 3DGS [16] and SuperGaussian [33] using their official implementations. Despite using fewer Gaussians, our method achieves significantly better NVS performance, with at least a **5.22** dB gain in PSNR. This demonstrates the superior capability of our method in modeling view-dependent color compared to previous works.

**Qualitative Results.** Fig. 2 shows the qualitative comparison between our method, 3DGS [16] and SuperGauss [33]. The results in the first two scenes reveal that 3DGS and SuperGauss struggle in reconstructing high-frequency details, while our method delivers more photorealistic results. We also show the two color components of our method, including a base image and a predicted residual map. While the base image alone exhibits the same detail deficiencies as 3DGS, adding our predicted color residuals helps to restore realistic textures. For scenes with complex view-dependent color, our method also demonstrates more compelling visual results, as shown in the last two scenes of Fig. 2. In these cases, 3DGS [16] and SuperGaussian [4] fail to capture the specular highlights and reflection effects, while our method successfully recovers the complex view-dependent colors in the final rendered images by predicting high residuals for these challenging regions. Interestingly, in the zoomed-in region of the Guitars scene, our method can decompose the color into a diffuse and specular component modeled via the base color and predicted residuals, respectively. Comparisons with more baseline methods and additional qualitative results can be found in the supplementary material.

## 4.3 Ablation Studies

Tab. 3 presents the ablation study results. **Base color only:** Discarding the predicted color residual leads to a significant drop in the image quality. This highlights the importance of the color residual module. **W/o color consistency loss $\mathcal{L}_{\text{photo}}$:** Training without the loss $\mathcal{L}_{\text{photo}}$ results in less accurate projections onto the source views, thereby reducing the quality of the rendered images. **Use source colors $c_m$ as network's input:** Here, we use the full colors $c_m^{\text{warp}}$ obtained from the source images

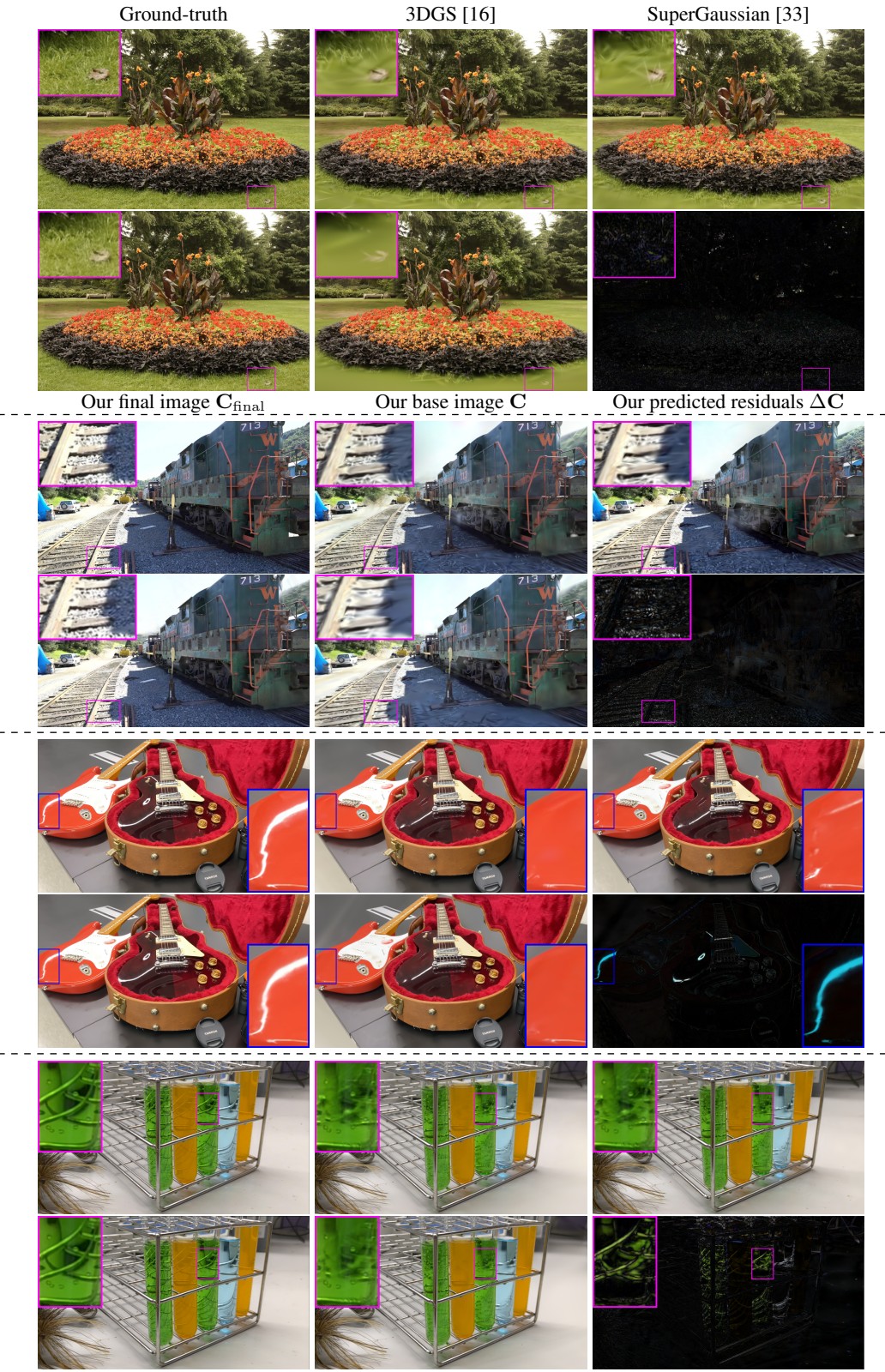

Figure 2: Qualitative results. Our method can render images with both high-frequency details (first two scenes) and view-dependent effect (last two scenes). However this cannot be achieved by 3DGS [16] and SuperGaussian [33]

Table 3: Ablation study on the exposure correction, color consistency loss and the network's input.

| Method | Tanks&Temples (TNT) | | | MipNeRF-360 | | |
|---|---|---|---|---|---|---|
| | PSNR↑ | SSIM↑ | LPIPS↓ | PSNR | SSIM | LPIPS |
| Full | **24.84** | **0.869** | **0.148** | **28.33** | **0.837** | **0.186** |
| Base color only | 23.06 | 0.836 | 0.202 | 27.08 | 0.814 | 0.227 |
| W/o color consistency loss $\mathcal{L}_{\text{photo}}$ | 24.70 | 0.866 | 0.152 | 28.31 | 0.833 | 0.192 |
| Use source color $\mathbf{c}_m^{\text{warp}}$ as network's input | 24.61 | 0.867 | 0.150 | 28.21 | **0.837** | 0.187 |
| W/o exposure correction | 24.28 | 0.866 | 0.152 | – | – | – |

Table 4: Ablation study on opacity threshold.

| Dataset | Mip-NeRF 360 | | | | | Tanks&Temples | | | | | Deep blending | | | | |
|---|---|---|---|---|---|---|---|---|---|---|---|---|---|---|---|
| Method (threshold) | PSNR↑ | SSIM↑ | LPIPS↓ | #Gauss | Mem | PSNR | SSIM | LPIPS | #Gauss | Mem | PSNR | SSIM | LPIPS | #Gauss | Mem |
| 3DGS (0.005) | 27.69 | 0.825 | 0.203 | 3.22 | 764 | 23.11 | 0.840 | 0.184 | 1.75 | 415 | 29.53 | 0.904 | 0.242 | 3.14 | 745 |
| Ours (0.005) | **28.42** | **0.836** | **0.183** | 2.61 | 456 | **24.76** | **0.869** | **0.146** | 1.23 | 220 | **29.91** | **0.907** | **0.234** | 2.41 | 405 |
| 3DGS (0.05) | 27.51 | 0.818 | 0.221 | 1.46 | 346 | 23.52 | 0.837 | 0.202 | 0.74 | 175 | 29.16 | 0.902 | 0.256 | 1.03 | 243 |
| Ours (0.05) | **28.33** | **0.837** | **0.186** | 1.59 | 291 | **24.84** | **0.869** | **0.148** | 0.75 | 143 | **30.12** | **0.912** | **0.237** | 1.11 | 197 |

as input to the residual prediction network, instead of their difference $\Delta\mathbf{c}_m$ from the base color. As a result, this approach performs consistently worse than our full model. **W/o exposure correction:** Removing the exposure correction also results in performance drop in the TNT dataset which exhibits inconsistent camera exposure across viewpoints [2]. Fig. 3 illustrates that our method can improve the exposure of the rendered image in both underexposure and overexposure cases, leading to higher image quality. More ablation studies can be found in the supplementary materials.

Additionally, we compare the sensitivity of our method and 3DGS [17] to the total number of Gaussians. For this experiment, we train our method and 3DGS with different opacity thresholds (0.005 and 0.05) used for pruning the Gaussians. Tab. 4 reveals that 3DGS requires a large number of low-opacity Gaussians to achieve good rendering quality, as its performance degrades when a higher opacity threshold is used. In contrast, with the same large threshold, our method can reduce the number of Gaussians while retaining most of the image quality.

## 5 Conclusion

In this paper, we present IBGS, an image-based Gaussian Splatting pipeline that is capable of rendering photorealistic images with both high-frequency details and view-dependent effects. Our key contribution is the color residual module, which leverages fine-grained textures and view-dependent information in nearby source images to predict a residual term added to the base Gaussian-rasterized color. Moreover, we introduce the exposure correction module to improve the brightness of the rendered image by mimicking the exposure of the closest source view. Extensive experimental results show that our method consistently outperforms previous works across different datasets.

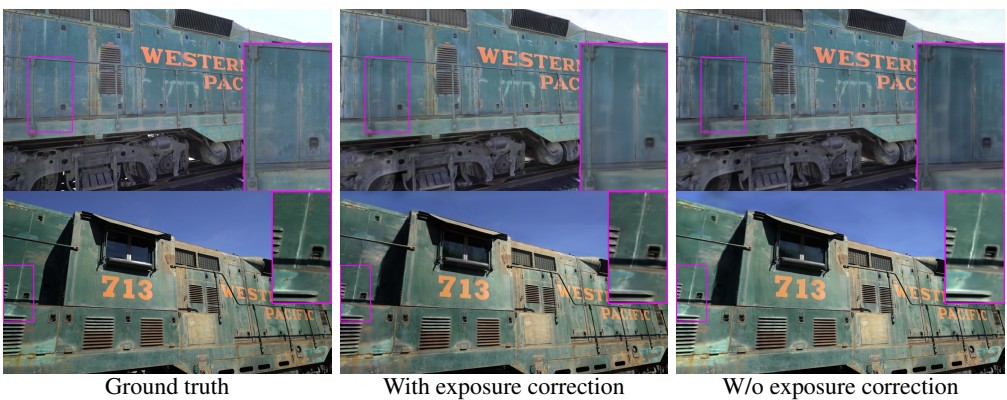

| Ground truth | With exposure correction | W/o exposure correction |

Figure 3: Comparison of rendered images with and without exposure correction. Our method can correct the exposure in both underexposure (top) and overexposure (bottom) cases.

**Limitations.** Our method may struggle in a sparse-view setting, in which obtaining dense pixel correspondences used for residual prediction is challenging. Additionally, due to the additional computations in the rendering process, our method achieves lower rendering speed and requires higher runtime memory compared to 3DGS [16]. We discuss this in more detail in the supplementary material.

**Broader Impacts.** Our method has no immediate societal impacts. However, its downstream applications, such as 3D reconstruction [15] or controllable human modeling [21], can potentially be abused for malicious purposes such as unauthorized reconstructions, identity fraud.

**Acknowledgement.** This research was supported in part by the Australia Research Council ARC Discovery Grant (DP200102274).

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
