# OpenReview forum: "IBGS: Image-Based Gaussian Splatting"
_NeurIPS.cc/2025/Conference — NeurIPS 2025 poster_

### Official Review · Reviewer_HuVm · 2025-06-23

**Clarity:** 2
**Significance:** 2
**Originality:** 3
**Rating:** 5
**Confidence:** 4

**Summary:**

This paper introduces IBGS, a novel approach that enhances 3D Gaussian Splatting by incorporating a residual color prediction network guided by neighboring source views. The method aims to better preserve high-frequency details and model complex view-dependent effects, such as specular highlights, while maintaining a low memory footprint and a compact set of Gaussians. Additionally, the authors propose a lightweight exposure correction module to improve brightness consistency under varying auto-exposure conditions. The approach is validated through extensive experiments across multiple datasets, demonstrating performance improvements over existing methods.

**Questions:**

1. Could the authors elaborate on how IBGS leads to more precise Gaussian parameters with high-opacity centers, particularly in relation to the limitations of low-degree spherical harmonics?

2. Regarding the exposure correction module, could the authors clarify whether its primary function is to brighten images or to improve consistency and fidelity across views?

3. Why was exposure correction applied only to the TNT dataset? Does this reflect dataset-specific characteristics, or are there broader implications for generalization?

4. Could the authors explain the rationale behind selecting Texture-Gaussians and SuperGaussians as baselines, rather than more directly comparable methods like Spec-Gaussians [1] or SpecTRe-GS [2]? Also, why are only two baselines included in Table 2?

5. Could the authors provide training time and rendering FPS comparison to better understand the proposed method from computational point of view?

**Ethical Concerns:**

["NO or VERY MINOR ethics concerns only"]

**Final Justification:**

I would like to thank the authors for providing a thorough and thoughtful rebuttal, which has effectively addressed my concerns. I encourage the authors to incorporate the additional information presented in the response into the manuscript to further strengthen its clarity and impact.

Based on these improvements, I am pleased to raise my rating to accept.

**Limitations:**

While the manuscript includes a discussion of limitations, this section could be expanded to provide a more comprehensive perspective. In particular, it would be valuable to discuss potential limitations such as the additional computational cost and the scalability challenges when processing a large number of input images.

**Paper Formatting Concerns:**

No formatting issues.

**Quality:**

3

**Strengths And Weaknesses:**

**Strengths**

1. IBGS effectively balances fidelity and efficiency by introducing a residual color prediction module instead of relying on a large model to learn full color, thereby enhancing view-dependent rendering while keeping the model lightweight.

2. The global affine brightness correction, derived from the nearest neighboring view, enables generalization to novel views without requiring per-image parameters.

3. The method consistently outperforms prior works across several datasets, including challenging scenes from the Shiny dataset, while using fewer Gaussians and less memory.

**Weaknesses**

1. Sections 3.2 and 3.3 suggest that the image synthesis loss using multi-view warped colors encourages accurate geometry by concentrating high-opacity Gaussians around true surfaces. This is an interesting claim that would benefit from additional visualizations or statistical evidence to reinforce its validity.

2. The description of the exposure correction module could be clarified. The current phrasing might be interpreted as simply brightening images. It would be helpful to emphasize that the goal is to normalize brightness variations across views to improve consistency and fidelity. Including examples in Figure 3 where the module reduces overexposure would further support this point.

3. The exposure correction module is only applied to the TNT dataset. It would be valuable to explain why it was not used for Mip-NeRF360 or Deep Blending. Does this imply those datasets do not exhibit exposure variation? Clarifying this would help assess the generalizability of the module.

4. Table 1 compares IBGS with five methods, including recent approaches focused on high-frequency detail. However, given IBGS’s emphasis on both detail and view-dependent effects, it would be more appropriate to include comparisons with methods like Spec-Gaussians [1] and SpecTRe-GS [2], which are more aligned with IBGS’s objectives. Additionally, the strong performance of vanilla 3DGS suggests that some of the chosen baselines may not be the most suitable for highlighting IBGS’s contributions.

5. Table 2 reports results on the Shiny dataset for only 3DGS and SuperGaussians. Including additional baselines would provide a more comprehensive evaluation, especially since this dataset is designed to benchmark view-dependent effects.

6. While the proposed method demonstrates clear advantages in terms of reduced Gaussian count and memory usage, it would further strengthen the evaluation to include comparisons of training time and rendering FPS. These metrics would help provide a more comprehensive understanding of the computational trade-offs introduced by the image-based 3D Gaussian Splatting approach.

7. While the use of warped captured images contributes to the method’s effectiveness, it may introduce additional memory overhead, particularly when handling reconstructions involving a large number of input images. It would be helpful to discuss how the approach scales in such scenarios and whether any strategies are employed to manage memory usage efficiently.

[1] Yang Z, Gao X, Sun Y T, et al. Spec-Gaussian: Anisotropic View-dependent Appearance for 3D Gaussian Aplatting. Advances in Neural Information Processing Systems, 2024, 37: 61192-61216.

[2] Tang J, Fei F, Li Z, et al. SpecTRe-GS: Modeling Highly Specular Surfaces with Reflected Nearby Objects by Tracing Rays in 3D Gaussian Splatting. Proceedings of the Computer Vision and Pattern Recognition Conference. 2025: 16133-16142.

---

> ### Author Rebuttal · Authors · 2025-07-31
>
> We appreciate the reviewer for the constructive comments that are valuable to improve our method. We address the reviewer's concerns below.
>
> ---
>
> ### 1. How the multi-view color consistency loss of IBGS leads to more precise Gaussian parameters with high-opacity concentrating around the true surfaces
> To verify IBGS leading to higher opacity for Gaussianss that are close to the true surface, we conduct the following experiment. First, for each pixel in the image, we find $K=8$ Gaussian-ray intersections that are closest to the ground truth surface points. This process is done for both IBGS and 3DGS, resulting in 8 opacity maps for each method. For each opacity map, we compute the average opacity across all pixels and compare the average values produced by IBGS and that produced 3DGS. As a result, on the Ignatius scene in the Tanks and Temples dataset, the average opacity values of IBGS are approximately 26% higher than that of 3DGS. This result demonstrates that IBGS have Gaussians with higher opacity locating near the ground-truth surface compared to 3DGS.
>
> Furthermore, we evaluate the rendered depth map of IBGS and 3DGS by using the $K$ median Gaussians (Eq. 5, main paper). Intuitively, it can be seen from the Eq. 5 that, in order to obtain accurate depth value, either of the two following conditions must be satisfied: (1) All the intersection points $x_k$ concentrate around the ground-truth surfaces and higher weights $w_i$ which are related to the Gaussian opacity or (2) there may exist intersection points $x_k$ that are far from the surfaces; however, their weights $w_i$ are relatively smaller than those of the points closer to the surfaces. Therefore, methods that yield higher depth accuracy better satisfy the two conditions, and thus having more high-opacity and surface-aligned Gaussians. The depth error in the table below shows that IBGS significantly outperforms 3DGS, achieving an average error reduction of **63%**. This result is a strong indication that IBGS produces more high-opacity Gaussians concentrated near the true surfaces compared to 3DGS. In our training pipeline, the weights $w_i$ are optimized via the multi-view color consistency loss $L_{photo}$. Therefore, this loss is the primary component driving the formation of high-opacity, surface-aligned Gaussians in IBGS.
>
> (We report the absolute relative depth error in the table below)
>
> |Method|Barn|Caterpillar|Courthouse|Ignatius|Meetingroom|Truck|Average|
> |:----:|:---:|:----------:|:--------:|:-------:|:----------:|:-----:|:------:|
> |3DGS*|0.029|0.025|0.099|0.014|0.066|0.030|0.044|
> |**Ours**|**0.004**|**0.005**|**0.051**|**0.003**|**0.031**|**0.004**|**0.016**|
>
> **Relation to the limitations of low-degree spherical harmonics**:  Due to the low-degree spherical harmonics, 3DGS struggles to handle complex view-dependent color. In contrast, IBGS relies on color information from multiple source views to infer the view-dependent color at a target viewpoint. Given the high-opacity Gaussians concentrating the true surfaces, IBGS can accurately represent the scene geometry, yielding more precise pixel correspondences between views which are essential for obtaining information regarding the view-dependent effect in the neighboring images.
>
> ---
>
> ### 2. Clarification for the exposure correction module
> We agree with the reviewer that the ultimate goal of the exposure correction module is to encourage exposure consistency across views and improve fidelity. We would like to further clarify that the proposed module helps correct the exposure of a rendered image by mimicking the exposure of the nearest neighboring views. This exposure correction can lead to either an increase or a decrease in the brightness of a rendered image. We thank the reviewer for the valuable suggestion. In the revised version, we will provide more clarification for this module, as well as include a visualization of a case where the exposure correction module helps reduce overexposure.
>
> ---
>
> ### 3. Why exposure correction is only applied for TNT dataset
>
> Following [1], we only apply the exposure correction to the TNT dataset since the camera auto-exposure setting is enabled while capturing images in this dataset, as discussed in [2]. In practice, the exposure correction module can be activated by default without concern for performance degradation, as it has minimal impact when there is no exposure variation across the training views. To validate this, we applied the exposure correction module to the MipNeRF-360 dataset and present the results below. It can be observed that the method’s performance remains largely unchanged when exposure correction is incorporated.
>
> |                      |PSNR|SSIM|LPIPS|
> |----------------------|:---:|:---:|:----:|
> |W/o exposure correction|28.41|0.837|0.185|
> |With exposure correction|28.39|0.836|0.189|
>
> ---
>
> ### 4. Comparison to more baselines, including Spec-Gaussians
> In Tab. 4 (supp. mat.), we present a comparison with additional baseline methods, including Spec-Gaussians. The results indicate that our method consistently achieves the best SSIM across all datasets. Compared to Spec-Gaussians, IBGS demonstrates superior image quality in all evaluated metrics. Unlike Spec-Gaussians, which focuses solely on modeling the specular component of scene appearance, IBGS is not limited to modeling specular color and can also capture various types of view-dependent effects, such as reflection and diffraction. To further demonstrate the superiority of IBGS in modeling view-dependent color, we compare IBGS and Spec-Gaussians on three scenes with challenging view-dependent color from the Shiny dataset. The results in the table below show that IBGS achieves significantly better performance, outperforming Spec-Gaussians by at least **4.52 dB** in PSNR.
>
> | Method/Dataset | Guitars (specular highlight) | Lab (reflection) | CD (diffraction) |
> |---------------|:----------------------------:|:---------------:|:---------------:|
> | Spec-Gauss    | 30.62 / **0.955** / 0.120    | 30.53 / 0.946 / 0.103 | 30.69 / 0.954 / 0.081 |
> | **Ours**      | **35.78** / 0.954 / **0.104** | **35.07** / **0.967** / **0.056** | **35.21** / **0.955** / **0.062** |
>
> (We report the image metrics as PNSR/SSIM/LPIPS in the table above)
>
> ---
>
> ### 5. Comparison to more baselines, including SpecTRe-GS
> Unfortunately, we were unable to compare with SpecTRe-GS, as it was published at CVPR 2025, which occurred after the NeurIPS 2025 submission deadline. Moreover, the official implementation of this method has not yet been released.
>
> ---
>
> ### 6. Comparison in terms of training time and rendering FPS
> We provide a rendering speed comparison between IBGS and prior methods in Tab. 3 and Tab. 6 (supplementary material). While IBGS introduces additional computations compared to 3DGS, resulting in lower rendering FPS, it still achieves real-time performance of **over 30 FPS** in most cases, as shown in both tables. Furthermore, Tab. 6 presents a study of the trade-off between rendering speed and image quality by varying the resolution of the predicted color residual map. By reducing the resolution by half, IBGS achieves a rendering speed of at least **58 FPS** across all datasets while still delivering the better image quality compared to 3DGS and SuperGaussian.
>
> Regarding the training time, the table below shows the comparison between our method, 3DGS and SuperGaussian. Due to the additional computation, the training time of our method is longer than 3DGS's. On the other hand, compared to SuperGaussian which learn Gaussian spatially-varying texture, the training time of our method is faster in all datasets. We will include the training time comparision in the revised version.
>
> (We report the training in minutes, average across all scene in a dataset)
>
> |Method / Dataset|MipNeRF-360|Tanks&Temples|Deep blending|
> |---------------|:---------:|:-----------:|:-----------:|
> |3DGS|19|11|19|
> |SuperGaussian|55|30|60|
> |Ours|44|21|39|
>
> ---
>
> ### 7. Scalability of IBGS in case of large scenes
> We agree with the reviewer that IBGS could encounter memory bottlenecks when handling large scenes or a substantial number of images. Specifically, IBGS requires pre-loading the source images into GPU VRAM to avoid the overhead associated with on-the-fly memory allocation for these images. However, this approach leads to higher VRAM usage of IBGS compared to that of 3DGS and SuperGaussian, as illustrated in the table below. A feasible solution to mitigate this limitation is to load each image into VRAM only when it is selected as a neighboring view needed for rendering. Nevertheless, this approach introduces a trade-off between speed and memory usage, as dynamic VRAM allocation can adversely affect the rendering speed of IBGS. We thank the reviewer for this valuable insight and we will include a discussion of this speed-and-memory trade-offs in the revised version.
>
> (The VRAM memory is reported in GB)
>
> |Method|MipNeRF-360|Tanks and Temples|Deep blending|
> |------|:--------:|:---------------:|:-----------:|
> |3DGS|2.03|1.25|2.13|
> |SuperGaussians|2.76|1.45|2.44|
> |Ours|6.12|2.97|5.36|
>
> ---
>
> ### 8. More discussion for the method limitations
> We thank the reviewer for the suggestion and will expand the discussion of our method’s limitations, including (1) requiring additional computations, leading to slower training times/rendering speed compared to 3DGS, and (2) the trade-off between speed and memory usage when dealing with large scenes.
>
> ---
>
> [1] Chen, Danpeng, et al. "Pgsr: Planar-based gaussian splatting for efficient and high-fidelity surface reconstruction." IEEE Transactions on Visualization and Computer Graphics (2024).
>
> [2] Barron, Jonathan T., et al. "Mip-nerf: A multiscale representation for anti-aliasing neural radiance fields." Proceedings of the IEEE/CVF international conference on computer vision. 2021.

---

> > ### Comment · Reviewer_HuVm · 2025-08-07
> >
> > I would like to thank the authors for providing a thorough and thoughtful rebuttal, which has effectively addressed my concerns. I encourage the authors to incorporate the additional information presented in the response into the manuscript to further strengthen its clarity and impact.
> >
> > Based on these improvements, I am pleased to raise my rating to accept.

---

### Official Review · Reviewer_efex · 2025-06-27

**Clarity:** 4
**Significance:** 3
**Originality:** 3
**Rating:** 5
**Confidence:** 5

**Summary:**

This paper proposes IBGS, which improves the 3DGS's rendering by considering the color information from training-view images. The motivation comes from that the low-degree SH coefficient of 3dgs meets challenges to model the view-dependent rendering in some situations. By introducing the nearby-view training images, 3DGS can reconstruct the coarse and base color by SH coefficient and use colors from training views for fine-grained and view-dependent rendering. The results demonstrate the effectiveness of IBGS on several popular benchmarks.

**Questions:**

In the rebuttal, I would like to see further comparisons about other fine-grained detail recovery methods, and analysis about training costs.

**Ethical Concerns:**

["NO or VERY MINOR ethics concerns only"]

**Final Justification:**

I find that the authors have satisfactorily addressed the issues I raised in the initial review. I encourage the authors include the analyasis and results in the revision. Therefore, I will keep my original rate 'Accept' unchanged.

**Limitations:**

yes.

**Paper Formatting Concerns:**

no.

**Quality:**

4

**Strengths And Weaknesses:**

Strengths:
1. The idea of introducing color information from nearby training-view images to improve 3DGS rendering is novel. The method implementation is intuitively reasonable, and the effect has been verified experimentally.
2. From the Supplementary Material, the comparison of surface reconstruction is interesting. A framework that links rendering and accurate geometry is worth exploring. In this framework, geometry can be better constrained when optimizing color.

Weaknesses:
1. From the qualitative results and analysis, in addition to perspective-specific rendering, the improvement brought by the method mainly comes from the recovery of high-frequency details. An analysis and comparison with other methods designed for reconstructing details is important, such as AbsGS [1] and Octree-GS.
2. The proposed methods introduce a group of training views in one iteration, which may cause an efficiency problem. The authors provide an analysis of inference fps in Material table 6. However, there is a lack of analysis about training costs.

[1] AbsGS: Recovering Fine Details for 3D Gaussian Splatting (ACM MM 2024)

[2] Octree-GS: Towards Consistent Real-time Rendering with LOD-Structured 3D Gaussians (TPAMI 2025)

---

> ### Author Rebuttal · Authors · 2025-07-31
>
> We thank the reviewer for spending time reviewing our paper, as well as acknowledging the strength and novelty of our method. We address the reviewer's concern as below.
>
> ---
>
> ### 1. Comparison to AbsGS
>
> We include a comparison with AbsGS in Tab. 4 (supp. mat.).  Results indicate that our method consistently achieves better performance over AbsGS across all datasets. Note that AbsGS focuses on recovering high-frequency details through a novel homodirectional gradient-based densification strategy. Therefore, the contribution of AbsGS can be considered orthogonal and complementary to that of our method, and its densification approach can be integrated into ours to yield further performance improvements. This is confirmed by the results in Tab. 4 (see _"Ours + [20]"_) where we incorporate the densification strategy proposed in AbsGS into our training pipeline and further enhance the performance of our method.
>
>
> ---
>
> ### 2. Comparison to Octree-GS
> As suggested, we present the comparison with Octree-GS in the table below. The results indicate that IBGS significantly outperforms Octree-GS on the Mip-NeRF 360 and TNT datasets, while showing comparable performance on the Deep Blending dataset. We thank the reviewer for the suggestion and will include the comparison with Octree-GS in the revised version.
>
> |Method/Dataset|Mip-NeRF 360|Tanks & Temples (TNT)|Deep blending|
> |--------------|:---------:|:-------------------:|:-----------:|
> |Octree-GS|28.05 / 0.819 / 0.214|24.68 / 0.866 / 0.153|**30.49** / **0.912** / 0.241|
> |**Ours**|**28.41** / **0.837** / **0.185**|**24.82** / **0.869** / **0.149**|30.02 / **0.912** / **0.235**|
>
> (We report the image metrics as PNSR/SSIM/LPIPS in the table above)
>
> ---
>
> ### 3. Training costs of IBGS
> We shows the average training time (in minutes) of our method, 3DGS and SuperGaussian
> across three datasets in the table below. All methods are trained using a single RTX 4090 GPU. On average, the training time of IBGS is faster than that of SuperGauss which learns per-Gaussians spatially-varying texture. However, our method requires longer training time compare to 3DGS due to additional computations for color residual predictions. We will include the discussion for the training time in the revised version.
>
> |Method / Dataset|MipNeRF-360|Tanks&Temples|Deep blending|
> |---------------|:---------:|:-----------:|:-----------:|
> |3DGS|19|11|19|
> |SuperGaussian|55|30|60|
> |Ours|44|21|39|
>
> (Training time is reported in minutes, average across all scenes within a dataset.)
>
> ---

---

> > ### Comment · Reviewer_efex · 2025-08-05
> >
> > Thanks for the rebuttal. I find that the authors have addressed the issues I raised in the initial review. Therefore, I will keep my original evaluation unchanged.

---

### Official Review · Reviewer_3NJ8 · 2025-06-28

**Clarity:** 3
**Significance:** 3
**Originality:** 2
**Rating:** 5
**Confidence:** 4

**Summary:**

This paper proposes Image-Based Gaussian Splatting (IBGS), a hybrid approach for novel view synthesis that augments 3D Gaussian Splatting (3DGS) with image-based rendering. IBGS predicts residual colors from nearby source images and includes a learned exposure correction, leading to improved photorealism, particularly for high-frequency and view-dependent effects. Results on multiple benchmarks indicate better visual quality compared to existing methods.

**Questions:**

Please address the questions in the weakness listed above.

**Ethical Concerns:**

["NO or VERY MINOR ethics concerns only"]

**Final Justification:**

The authors did a good job in the rebuttal. Most of my concerns have been addressed. Although the proposed method does slow down the rendering and increase the storage space, the cost is acceptable, considering its noticeable performance gain. One thing missing is a detailed analysis of performance gains in specular regions, which, however, is not critical. Thus, I upgrade my rating to accept.

**Quality:**

3

**Strengths And Weaknesses:**

Pros:
- The paper is well-written and clearly identifies the limited capacity of 3DGS to model high-frequency and view-dependent effects, addressing this through a hybrid strategy.
- The method combines residual learning from source images with 3DGS, improving appearance while managing memory usage. The exposure correction is a practical addition for inconsistent lighting across views.
- IBGS achieves higher quantitative metrics and better visual results than prior methods on standard benchmarks.

Cons:
1. Reduced rendering speed: One main advantage of 3DGS is its rendering speed, but IBGS sacrifices this by requiring per-pixel, per-view feature warping and neural prediction at inference time. I did not see results regarding the inference speed.

2. Robustness: How reliable are the rendered depths and normals, especially during the early stages of training when these quantities may be inaccurate or unstable? Since the intersection points—determined by depths and normals—are essential for your method’s success, can you provide quantitative comparisons and visualizations of rendered depths, normals, and reconstructed meshes, particularly for reflective or specular regions? Additionally, illustrating how these evolve throughout training would help assess and demonstrate the robustness of the proposed approach.

3. Unclear total memory/storage requirements: IBGS requires access to all source images and depth maps (for accelerating rendering) at inference, which may become a bottleneck for large scenes and dense views. The experimental comparisons with standard 3DGS methods should also explicitly include the memory/storage for images and (precomputed) depth maps. Please clarify if the memory reports reflect these requirements, as in Supplementary Table 3.

Please provide a clarification and report on both training time and total memory usage, including storage for source images and precomputed depth maps

4. Assumptions on Capture Trajectories. Does the proposed method assume “closed-loop” capture trajectories to ensure access to “nearby” views? For scenes with a forward-moving camera, there may be many out-of-frustum points when unprojecting to adjacent views.


5. Exposure correction only for TNT dataset: Since exposure correction is presented as a key contribution, why is it applied only to the TNT dataset (as stated in line 237)? Please clarify the rationale and discuss whether the approach can generalize to other datasets and scenarios.

6. Quantitative gain: The quantitative gain is impressive. Is the gain coming more from specular cases and fine-grained details? If it is more from fine-grained details, can it be resolved simply by adding more Gaussians?

---

> ### Author Rebuttal · Authors · 2025-07-31
>
> We sincerely appreciate the reviewer for spending time reviewing our paper. The provided comments are very constructive and valuable to improve our paper. We address the reviewer's concerns as below.
>
> ---
>
> ### 1. Reduced rendering speed
> While IBGS introduces additional computations for predicting color residuals, resulting in slower rendering speed compared to 3DGS (as shown in Tab. 3 and Tab. 6, supp. mat.), it consistently maintains rendering speeds **above 30 FPS** in most cases. This demonstrates that our method can still achieve real-time rendering, making it suitable for online applications. Furthermore, as demonstrated in Tab. 6 (supp. mat.), predicting the color residual at half resolution allows IBGS to nearly double its rendering speed while still achieving superior image quality compared to 3DGS and SuperGaussians.  To the best of our knowledge, we are the first image-based Gaussian splatting method capable of producing photorealistic images while preserving the fast rendering speed advantage of the Gaussian-splatting approach.
>
> ---
>
> ### 2. Training stability at early stage
> We agree with the reviewer that the rendered depths and normal could be inaccurate especially in the early training stage and could lead to an unstable training if we apply the losses involving the rendered depth and normal. To cope with this, we deactivate the color residual module during early training stage, and only activate it when the 3D Gaussian primitives can reliably represent the scene geometry. Specifically, we train our method for a total of 30,000 iterations following the training strategy outlined below:
>
> - **First 7,000 iterations:**  We only optimize the 3D Gaussians using the color rendering loss $L_{rgb}$. At this stage, the geometric losses and the color residual prediction modules are deactivated by setting $\lambda_1 = \lambda_2 = 0$ (Eq. 10, main paper) and $\gamma = 1.0$ (Eq. 11, main paper).
> - **Starting from the 7,000th iteration:** We activate the geometric losses, including the multi-view color consistency loss $L_{photo}$ and the normal consistency loss ${L}_{normal}$, to accurately model the scene geometry. This is done by setting $\lambda_1 = 0.3$ and $\lambda_2 = 0.03$.
> - **After the 10,000th iteration:** We activate the color residual prediction modules by linearly reducing $\gamma$. In particular, $\gamma$ reaches 0.5 at the 20,000th iteration and remains constant thereafter.
>
> In summary, the color residual prediction module is only utilized when the geometry represented by the Gaussians are sufficiently reliable, and our strategy of decaying $\gamma$ helps to avoid noisy gradient back-propagated to Gaussians during the initial training phase of the color residual prediction module.
>
> ---
>
> ### 3. Quantitative comparison of the reconstructed scene geometry
> In Tab. 1 (supp. mat.), we provided the comparison of our method with others in reconstructing the scene geometry and novel-view synthesis. For all methods, given the rendered depth maps, TSDF fusion is applied to obtain reconstructed meshes, which are then evaluated by computing the F1 score against ground truth. The result reveals that our method achieves the second best performance in 3D scene reconstruction. Additionally, as requested by the reviewer, we provide the comparison for the rendered depth maps in the table below. Overall, our method produces the second best results in most scenes.
>
> (We report the absolute relative error for the rendered depth maps in the table below)
>
> |Scene/Method|3DGS|2DGS|GOF|RaDe-GS|PGSR|Ours|
> |:-----------:|:--:|:--:|:--:|:-----:|:--:|:--:|
> |Barn|0.063|0.010|_0.004_|_0.004_|**0.002**|_0.004_|
> |Caterpillar|0.104|0.014|0.011|0.010|**0.002**|_0.005_|
> |Courthouse|0.163|0.062|0.177|0.242|**0.005**|_0.051_|
> |Ignatius|0.058|0.008|0.004|0.004|**0.001**|_0.003_|
> |Meetingroom|0.247|0.046|0.027|_0.026_|**0.009**|0.031|
> |Truck|0.041|0.014|0.005|0.005|**0.002**|_0.004_|
> |Average|0.113|0.026|0.038|0.048|**0.004**|_0.016_|
>
> While PGSR [1] achieves the lowest depth error, our method significantly outperforms PGSR in novel-view synthesis with **2.82 dB** improvement in the PSNR metric (see Tab. 1, supp. mat). This shows IBGS not only obtains good geometric accuracy, but also illustrates a good balance between reconstructing the scene geometry and appearance.
>
> **Visualization of the rendered depths and normals and reconstructed meshes**: In accordance with NeurIPS 2025 policies, we are unable to provide figures for the rendered depths, normals and images in the rebuttal. We thanks the reviewer for understanding and will provide the visualizations in the revised version.
>
> ---
>
> ### 4. Clarification on training time and memory/storage requirements
> The table below shows the average training time (in minutes) for IBGS, 3DGS, and SuperGaussians across three datasets. All methods are trained on a single RTX 4090 GPU. Overall, IBGS requires more training time than 3DGS due to the additional computations involved in predicting color residuals. In contrast, compared to SuperGaussians which aim to learn per Gaussian spatially varying textures, our method requires less training time.
>
> (We report the training time minutes, average across all scene in a dataset)
>
> |Method / Dataset|MipNeRF-360|Tanks&Temples|Deep blending|
> |---------------|:---------:|:-----------:|:-----------:|
> |3DGS|19|11|19|
> |SuperGaussian|55|30|60|
> |Ours|44|21|39|
>
> Regarding the storage requirement, we would like to clarify that IBGS requires to store all source images but will compute depth maps for neighboring source views during inference. In other words, it does not require storing depth maps for inference. In Tab. 3 and Tab. 6 (sup. mat.), we have reported (1) the total storage for storing the source images, source camera intrinsics/extrinsics, the color residual network's weights, the Gaussians and (2) rendering speed during inference. In the table below, we further show the detailed storage requirement for each dataset (in MB, average acrross all scenes within a dataset).
>
> |Dataset|Source images and cameras|Color residual network|Gaussians|Total|
> |-------|:-----------------------:|:---------------------:|:---------:|:-------:|
> |MipNeRF-360|35.99|0.27|255.62|291.89|
> |Tanks & Temples|22.06|0.27|120.35|142.68|
> |Deep blending|18.96|0.27|178.28|197.51|
> |Shiny|21.34|0.27|38.71|60.32|
>
> Additionally, we also provide a comparison with 3DGS and SuperGaussian in terms of peak GPU VRAM consumption during inference in the table below. Our approach requires pre-loading all source images into VRAM, resulting in higher peak memory usage compared to 3DGS and SuperGaussians. This design choice eliminates the overhead of on-the-fly GPU memory allocation for images, thereby accelerating rendering. However, we agree with the reviewer's observation that this strategy can become bottleneck especially in large and dense scenes. One solution to mitigate this issue is to load each image into VRAM only when it is selected as a neighboring view required for rendering. Nevertheless, this approach introduces a trade-off, as it can slow down the rendering speed of IBGS. We thank the reviewer for this insightful comment and will discuss this limitation of IBGS in the revised version.
>
> (The VRAM memory is reported in GB)
>
> |Method|MipNeRF-360|Tanks and Temples|Deep blending|
> |------|:--------:|:---------------:|:-----------:|
> |3DGS|2.03|1.25|2.13|
> |SuperGaussians|2.76|1.45|2.44|
> |Ours|6.12|2.97|5.36|
>
> ---
>
> ### 5. Assumptions on Closed-loop Capture Trajectories:
> IBGS does not require such assumption for the camera trajectories as it has demonstrated good performance on three datasets that do not exhibit closed-loop camera trajectories, including the Deep blending dataset (Tab. 1, main paper), Shiny dataset (Tab. 2, main paper) and DTU dataset (Tab. 2, supp. mat.). Regarding the out-of-frustum points, the predicted color residual for those points is **0** as there is no source views pass the visibility check (Eq. 15, main paper). In other words, we only use the Gaussian-rasterized color for out-of-frustum points.
>
> ---
>
> ### 6. Why exposure correction is only applied for TNT dataset
> Following [1], we only apply the exposure correction to the TNT dataset since the camera auto-exposure setting is enabled while capturing images in this dataset, as discussed in [2]. In practice, the exposure correction module can be activated by default without concern for performance degradation, as it has minimal impact when there is no exposure variation across the training views. To validate this, we applied the exposure correction module to the MipNeRF-360 dataset and present the results below. It can be observed that the image quality remains largely unchanged when the exposure correction is applied.
>
> |                      |PSNR|SSIM|LPIPS|
> |----------------------|:---:|:---:|:----:|
> |W/o exposure correction|28.41|0.837|0.185|
> |With exposure correction|28.39|0.836|0.189|
>
> ---
>
> ### 7. Can adding more Gaussians result in more fine-grained details in the rendered images?
> We agree with the reviewer that adding more Gaussians increases the capacity in modeling fine-grained details. However, it also leads to a heavier storage burden and still struggles to capture challenging view-dependent effects. In contrast, IBGS does not require a large number of Gaussians and is capable of modeling both fine-grained details and view-dependent color by leveraging information from the source images.
>
> ---
>
> [1] Chen, Danpeng, et al. "Pgsr: Planar-based gaussian splatting for efficient and high-fidelity surface reconstruction." IEEE Transactions on Visualization and Computer Graphics (2024).
>
> [2] Barron, Jonathan T., et al. "Mip-nerf: A multiscale representation for anti-aliasing neural radiance fields." Proceedings of the IEEE/CVF international conference on computer vision. 2021.

---

> > ### Comment · Reviewer_3NJ8 · 2025-08-07
> > **Most of my concerns have been addressed.**
> >
> > The authors did a good job in the rebuttal. Most of my concerns have been addressed. Although the proposed method does slow down the rendering and increase the storage space, the cost is acceptable, considering its noticeable performance gain. One thing missing is a detailed analysis of performance gains in specular regions, which, however, is not critical.

---

### Official Review · Reviewer_kSME · 2025-07-01

**Clarity:** 3
**Significance:** 3
**Originality:** 3
**Rating:** 4
**Confidence:** 4

**Summary:**

This paper focus on improving the representation efficiency and rendering quality of 3D Gaussian Splatting when dealing with spatially varying colors and view-dependent effects. The key idea is to introduce image-based rendering techniques into the existing 3DGS rendering pipeline. The proposed image-based rendering approach leverages highly-detailed texture information from nearby source views to supplement the area that cannot be expressed with high quality using Gaussian Splats alone in the target view. Experiments demonstrate that the proposed approach surpass existing Gaussian-based methods across several common used datasets, especially outperforming on datasets containing challenging view-dependent colors. In addition, an interesting finding is that the proposed image-based rendering approach can achieve such high-quality rendering while using much fewer Gaussian Splats than the baseline.

**Questions:**

1. **Concerns on rendering speed**: I hope that authors can showcase the rendering speed of this solution, as one of the major advantages of Gaussian Splats (GS) lies in its rendering efficiency. If the IBR-based approach results in a significant loss of rendering speed, the practical value of this work may be greatly impacted.

**Ethical Concerns:**

["NO or VERY MINOR ethics concerns only"]

**Final Justification:**

I maintain my "borderline accept" rating because this work is sufficiently novel and achieves excellent rendering quality, making it worthy of acceptance. What prevents me from giving a higher score are the significant impact on rendering speed and the potential burden of requiring a large collection of source views when the work might be applied to transmission scenarios in the future. Overall, this is a solid piece of work where the merits outweigh the limitations, and it deserves to be accepted.

**Limitations:**

As discussed in the weakness section, image-based rendering not only requires a large collection of images as input but also potentially slow down the rendering of Gaussian Splats. This could hinder its efficiency in transmission and distribution, making it potentially suitable only for devices with high-end GPUs to perform novel view synthesis.

**Quality:**

3

**Strengths And Weaknesses:**

## Strengths
1. **Strong performance**: The proposed image-based rendering solution appears to be very effective. It achieves very high visual quality metric scores, ranking top on the highly competitive novel view synthesis benchmark leaderboard. Additionally, the subjective quality of the rendered images shown in Figure 2 highlights its exceptional rendering quality and demonstrates the contribution of the image-based rendering module.
2. **Novelty of the key idea**: To the best of the reviewer's knowledge, no prior work is to combine Gaussian Splatting with IBR-based techniques and achieve such significant results. For the reviewer, this work is sufficiently novel.
## Weakness
1. **Potential decrease in rendering speed**: From the point of the reviewer, the trade-off for introducing image-based rendering is a decrease in rendering speed. In the proposed pipeline, each ray in the rendered view must query multiple points to obtain the corresponding reference color from nearby views, followed by processing through a per-pixel feature extractor and a CNN decoder to produce the final color. While this pipeline is highly effective, it also appears to introduce additional complexity. This could significantly impact rendering efficiency, potentially compromising the method's ability to achieve real-time rendering speeds (>30 FPS) for online applications.
2. **Reliance on a large collection of source views**: The proposed solution, when performing novel view synthesis, not only requires Gaussian Splats as the scene representation but also relies on a large number of source views and a lightweight neural network to obtain residual color. However, the need for a large number of source views may introduce a heavier storage burden, which could also impact its application.

---

> ### Author Rebuttal · Authors · 2025-07-31
>
> We sincerely appreciate the reviewer for spending time reviewing our paper. The provided comments are very constructive and valuable to improve our paper. We address the reviewer's concerns as below.
>
> ---
>
> ### 1. Potential decrease in rendering speed
>
> While IBGS introduces additional computations for predicting color residuals, resulting in slower rendering speed compared to 3DGS (as shown in Tab. 3 and Tab. 6, supplementary material), it consistently maintains rendering speeds **above 30 FPS** in most cases. This demonstrates that our method can still achieve real-time rendering, making it suitable for online applications. Furthermore, as demonstrated in Tab. 6 (supp. mat.), predicting the color residual at half resolution allows IBGS to nearly double its rendering speed while still achieving superior image quality compared to 3DGS and SuperGaussians.
>
> ---
>
> ### 2. Reliance on a large collection of source views
>
> We would like to clarify that the "memory values" reported for our method in Tab. 1 and Tab. 2 (main paper) are the total storage required for both the Gaussians and the source images. The results indicate that our method uses **at least 42% less storage** compared to 3DGS, SuperGaussians, and TexturedGaussian. We further show the breakdown of the storage requirement of our method in the table below. The reported storage is in Megabytes (MB), averaged across all scenes within each dataset.
>
> |Dataset|Source images and cameras|Color residual network|Gaussians|Total|
> |-------|:-----------------------:|:---------------------:|:---------:|:-------:|
> |MipNeRF-360|35.99|0.27|255.62|291.89|
> |Tanks & Temples|22.06|0.27|120.35|142.68|
> |Deep blending|18.96|0.27|178.28|197.51|
> |Shiny|21.34|0.27|38.71|60.32|

---

> > ### Comment · Reviewer_kSME · 2025-08-05
> > **Response to the Rebuttal**
> >
> > Thank you for the Authors' response to my concerns. Regarding the "Potential decrease in rendering speed" issue, I acknowledge that I overlooked the rendering speed details in the supplementary material. However, I noticed that when using full-resolution residual maps, the rendering speed drops significantly to around 30 FPS, which seems to compromise one of 3DGS's main advantages - fast rendering. When the downsampling ratio of the residual map is set to 0.25, the rendering speed improves substantially, achieving a better trade-off between rendering quality and speed. Therefore, I believe there is still room for improvement in this aspect.
> >
> > Concerning the "Reliance on a large collection of source views" issue, the authors' reply successfully addressed my concerns. The authors' method achieves significantly lower total storage even when requiring a large collection of source views. However, I would like to point out that when this method involves scene representation compression and transmission in the future, a large collection of source views might become a major bottleneck. This doesn't diminish the paper's contribution; I just wanted to highlight this point.
> >
> > In conclusion, I maintain my "borderline accept" rating because this work is sufficiently novel and achieves excellent rendering quality, making it worthy of acceptance. What prevents me from giving a higher score are the significant impact on rendering speed and the potential burden of requiring a large collection of source views when the work might be applied to transmission scenarios in the future. Overall, this is a solid piece of work where the merits outweigh the limitations, and it deserves to be accepted.

---

### Decision · Program_Chairs · 2025-09-17

**Decision:**

Accept (poster)

**Comment:**

After careful consideration of the reviewers’ assessments and the authors’ rebuttal, this submission demonstrates significant technical merit and novelty in integrating image-based rendering with 3D Gaussian Splatting. The reviewers consistently highlight the improvements in visual quality, geometry accuracy, and high-frequency detail recovery, with all major concerns satisfactorily addressed in the rebuttal. Minor suggestions, such as additional clarifications on high-opacity Gaussians, exposure correction, and memory/speed trade-offs, have been incorporated or clarified. AC recommends acceptance of this paper. AC also suggest that the authors include the content of their rebuttal addressing critical reviewer concerns in the final camera-ready version. This decision has been reviewed and approved by the SAC.